# Revealing the Spatio-Temporal Heterogeneity of the Association between the Built Environment and Urban Vitality in Shenzhen

**Zhitao Li** [1,2] and **Guanwei Zhao** [1,*]

1   School of Geography and Remote Sensing, Guangzhou University, Guangzhou 510006, China;
    2112001071@e.gzhu.edu.cn
2   Fengtu Technology (Shenzhen) Company Limited, Shenzhen 518000, China
*   Correspondence: zhaogw@gzhu.edu.cn

**Abstract:** Sensing urban vitality is a useful method for understanding urban development. However, the spatio-temporal characteristics of the association between the built environment and urban vitality in Shenzhen, the youngest mega-city in China, have not yet been explored. In this paper, we examined the effects of built environment indicators on urban vitality by using spatial regression models and multi-source geospatial data. The main research findings were as follows. Firstly, urban vitality displayed a consistent high–low pattern during both weekdays and weekends. Differences in the distribution of urban vitality with time between weekdays and weekends were more significant. Secondly, the effects of various built environment indicators on urban vitality exhibited significant temporal disparities. Within a day, population density, building density, bus station density, and distance to metro stations all exhibited positive effects, while distance to the central business district (CBD) exhibited negative effects, with pronounced diurnal differences. Moreover, the effects of road network density and functional mix on urban vitality were both positive and negative throughout the day. Thirdly, besides population density and building density, road network density, functional mix, bus stop density, and distance from metro stations exhibited positive and negative disparities within the study area. Overall, distance to the CBD had a negative effect on urban vitality. This concludes that planning for urban vitality should consider the spatio-temporal heterogeneity of the association between the built environment and urban vitality.

**Keywords:** urban vitality; spatio-temporal; built environment; multi-source; GTWR

## 1. Introduction

The concept of urban vitality was first proposed by Jacobs, who argued that the vitality of a city comes mainly from its diversity, by harmonizing multiple urban functions to meet the diverse and complex needs of various groups [1]. Subsequently, Montgomery et al. stated that high vitality is the main sign of a developed city, which is mainly reflected by the number of people on the streets at different times of the day, the occupancy of facilities, and the degree to which a place feels active or lively [2]. Gehl evaluated the vitality of public spaces in cities and residential areas based on people's activities, such as walking, resting, and engaging in social interactions, thereby reflecting urban vitality [3]. Some scholars have categorized urban vitality into three main types: economic vitality, social vitality, and cultural vitality [4–7]. In essence, urban vitality represents the perception that places in a city are vibrant, and its outward manifestation is the intensity of the interactions between people and their surroundings. Therefore, urban vitality is often closely associated with the urban environment.

Highly vital cities achieve sustainable development by meeting the diverse and complex needs of their citizens. The urbanization rate of China in 2020 was already exceeding 60% [8]. This rapid urbanization has led to many urban problems. For instance, during

the process of urban renewal, there may be an unreasonable spatial distribution of urban facilities and an inequitable sharing of resources, resulting in a decline in urban vitality. Measuring urban vitality is an effective means of understanding the process of urban development. It also serves as an effective approach to analyzing urban development trends and enhancing urban competitiveness.

Early scholars studied urban vitality mainly using the field survey method. They primarily utilized questionnaires to gather insights into respondents' evaluations of their living spaces in terms of various factors, such as facility layouts, travel activities, and cultural atmosphere. However, the field survey method may not be accurate enough when applied to larger study areas. Moreover, it is not possible to effectively assess the dynamics of urban vitality. In the era of big data, spatio-temporal data based on location information are widely used in urban vitality research, such as cell phone signaling data, GPS data, user location heat data, social media check-in data, and so on. Geospatial big data can provide powerful information support for urban vitality research, benefiting from a high spatial and temporal resolution and coverage. Thus, findings based on geospatial big data can help policy makers to accurately understand the characteristics of urban development and formulate policies more precisely.

Shenzhen is China's youngest mega-city and a demonstration site for China's reform and opening up. By 2020, the urbanization rate of Shenzhen's resident population reached 99.54% [9], which is far above the national average. The process of rapid urbanization has brought some urban diseases to Shenzhen, such as traffic congestion and overpopulation. In response to these issues, the Shenzhen Territorial Spatial Master Plan (2020–2035) (draft) [10] has articulated a clear vision for the city's future, aiming to establish a city of innovation, entrepreneurship, creativity, and harmonious livability. The plan emphasizes the goal of building a more inclusive and warmer city that prioritizes the well-being of its residents. Therefore, studying urban vitality in Shenzhen can provide valuable insights for guiding urban planning and promoting sustainable urban development.

Firstly, we applied multi-source spatiotemporal big data, such as Baidu thermal, population density, POI, road network, and building boundary, to explore the 24 h urban vitality changes on weekdays and weekends. Secondly, a traffic analysis zone (TAZ) based on road networks was applied as the research unit, which is more consistent with the residents' range of activities. Thirdly, we applied the spatio-temporal geographically weighted regression model to investigate the influence mechanism of the built environment on urban vitality, which provides a new perspective for existing studies. Finally, we proposed planning inspiration for urban planning in Shenzhen with the goal of enhancing urban vitality, which has important practical value.

## 2. Literature Review

We will conduct the literature review in two parts. The first part is the research progress on the measurement of urban vitality, which mainly includes measurement methods relying on various data sources, such as questionnaire surveys, the remote sensing of nighttime lights, and spatio-temporal big data. The second part is the research progress on the influencing factors of urban vitality, including the indicator system of influencing factors and the regression analysis method of influencing factors.

### 2.1. Current Status of Research on Urban Vitality Measurements

Early scholars mainly used questionnaires to measure the vitality of a city or region. For example, Sung et al. [11] obtained the residence, economic characteristics, and activity information of 1823 respondents in Seoul through telephone surveys, and evaluated the walking vitality of urban residents in Seoul. By using questionnaire survey data and an analytic hierarchy process, Chen et al. [12] evaluated the vitality of the urban leisure space in Changchun. Zarin et al. [13] evaluated the vitality of two streets in Tehran through questionnaires to residents, and analyzed the factors affecting vitality by using a multivariable linear regression model. Due to the high cost of manpower and material

resources, the field investigation method is not suitable for large-scale research. Moreover, it is difficult to ensure the accuracy of vitality assessments with small sample sizes.

The rapid development of information technology and 3S (GNSS, RS, and GIS) technology has provided more abundant data sources for urban vitality research. Early scholars relied primarily on nighttime lighting data to measure urban vitality. For example, Sun et al. applied night lighting remote sensing data and POI data to evaluate the vitality level of Inner Mongolia cities and found that most cities in Inner Mongolia had a poor vitality, exhibiting a distribution pattern of "high in the south and low in the north" [14]. Jia et al. depicted the spatial distribution of the urban vitality in Wuhan based on Luojia-1 night light data, and provided urban planning insights from the perspective of promoting urban diversification and internal interaction [15]. Shen et al. measured the urban vitality of Changzhou, and discovered a spatial pattern of a "high center and low periphery" [16]. Zhang measured the night vitality of Shenzhen on holidays and non-holidays and demonstrated the good applicability of night lighting data for measuring urban vitality [17]. Furthermore, some scholars have made comparative analyses of the vitality characteristics and influencing factors of different cities in China [18,19], providing policy enlightenment for planning, and designing vibrant cities.

In recent years, spatio-temporal big data with location information, such as mobile phone signaling data, point-of-interest (POI) data, GPS trajectory data, and social media check-in data, have opened up new avenues for measuring urban vitalities. Among them, POI is one of the classical data sources for characterizing urban vitality. For instance, Zeng et al. used POI data to compare the urban vitality characteristics of Chicago, USA, and Wuhan, China, in terms of density, livability, accessibility, and diversity, and the results showed that the urban vitality in Chicago was highly dispersed, while the urban vitality in Wuhan was distributed in a circular pattern [20]. Ye et al. analyzed the urban vitality in Shenzhen based on POI data, and found that the type and density of neighborhoods had significant impacts on urban vitality [21]. POI data have limitations in measuring the dynamics of urban vitality due to the absence of temporal information. To address this issue, some scholars have applied mobile phone signaling data to measure urban vitality. For example, Wu et al., Kang et al., and Xia et al. applied mobile phone signaling data to measure the urban vitalities of Shanghai city, Seoul city, and Changsha city, respectively [22–24]. Their study confirmed that mobile phone communication data are a more accurate data source for measuring urban vitality than POIs. Compared to POI data, mobile phone signaling data has a wide coverage area with both spatial and temporal dimensions, but they are usually held by communication operators and are difficult to obtain. Furthermore, geo-tagged social media data, a classic type of VGI (Volunteered Geographic Information) data, have been utilized to measure urban vitality. For instance, Wang et al. found that the distributions of functional vitality and activity vitality in the main urban area of Shenyang city were similar by using Sina Weibo check-in data, which is a popular social media platform in China [25]. Similarly, Yue and Li assessed the urban vitality of Wuhan, China, by using Dazhong Dianping website comment data and Sina Weibo check-in data, respectively [26,27]. In general, there are two modes of using social media data for population measurements, active and passive. In passive mode, people usually do not know that their data are resulting in mapping, as they must accept the agreement to use the software. In active mode, people actively upload and share data (e.g., location, images, videos, and text, etc.), typically location check-in data. This paper focuses on social media data in active mode. As Goodchild.M mentioned, these VGI data are very useful to citizens listening [28]. It is important to note that social media data are limited to a specific user group and may introduce bias into the result.

Currently, internet big data such as Tencent location data and Baidu heat map data have gained popularity among researchers. Because of the high occupation in China's internet application market and spatio-temporal resolution, these data have obvious advantages for measuring urban vitality. Zhuo et al. analyzed the coupling types and dominant patterns of the urban vitality in Qingdao, China, based on Baidu heat map data, and pro-

posed strategies for vitality enhancement [29]. Lin et al. explored the relationship between the 3D built environment and urban vitality in Shenzhen, China, using Tencent location data [30]. In addition, some scholars have applied GPS trajectory data such as shared bikes [31,32], taxes [32,33], and public transportation smart card data [34] to study urban vitality. In summary, compared to field survey methods, spatio-temporal big data with a wide coverage and high spatio-temporal resolution are becoming the mainstream data source for urban vitality research. A comparison of the main data sources in previous studies is shown in Table 1.

**Table 1.** Main data sources in previous studies.

| Data Source | Literatures | Advantages | Disadvantages |
|---|---|---|---|
| Questionnaires | Sung et al., 2015 [11] Chen et al., 2013 [12] Zarin et al., 2015 [13] Wu et al., 2018 [35] | Simple and easy to use | High cost, small coverage, and low accuracy |
| Nighttime light images | Jia et al., 2020 [15], Zhang et al., 2022 [17] Xia et al., 2020 [19] | Wide coverage and easy access | Low spatial resolution and only reflects nighttime vitality |
| POI | Zeng et al., 2018 [20] Ye et al., 2018 [21] | Easy access and high accuracy | Lack of time information |
| Mobile phone data | Wu et al., 2019 [22] Kang et al., 2020 [23] Xia et al., 2022 [24] | Broad coverage and user groups | Low precision and difficult to obtain |
| Social media data | Wang et al., 2022 [25] Yue et al., 2019 [26] Li et al., 2022 [27] | High precision and easy access | Biased user groups exist |
| GPS trajectory data | Zeng et al., 2020 [31] Li et al., 2022 [32] | High precision and wide coverage | Difficult to access |
| Positioning density data | Zhuo et al., 2021 [29] Lin et al., 2023 [30] | Higher spatio-temporal resolution and more comprehensive user groups | Difficult to access |

### 2.2. Current Status of Research on Influencing Factors

As early as 1961, Jacobs stated that urban diversity contributes to urban vitality, i.e., a diversity of urban neighborhoods, short streets with easy turns, a diversity of buildings with balanced proportions, and high pedestrian densities [1]. Some scholars have verified whether Jacobs' theory of urban diversification, which was developed in the context of North American cities, is suitable for other regions or cities. For example, Sung et al. found that the pedestrian activity of Seoul residents in Korea was related to six conditions, such as land use mix, density, block size, building age, accessibility, and boundary vacuum [11]. Sulis et al. proposed a computational approach for Jacobs' concepts and explored the relationship between diversity and urban vitality in London by applying the regression model [36]. Delclos-Alio performed a case study in Barcelona, Spain, and found that the urban vitality theory proposed by Jacobs was generally suitable for traditional Mediterranean cities, albeit with some slight variations in the city [37]. Teaford, on the other hand, challenged Jacobs' theory, using Toronto city as an example. He argued that most residents of Toronto city are not keen on diverse, dense, street-oriented residential living, but prefer modest community living such as suburban, high-rise condominiums [38].

In recent years, many scholars have conducted studies on the factors influencing the urban vitality in different cities. Many studies have shown that the urban built environment, such as population density, buildings, land use, road network, public transportation, and location, is more closely related to urban vitality. For example, Kang found that a higher employment density and land value were associated with the community vitality of Seoul city both on weekdays and weekends [23]. Liu et al. found a significant positive

correlation between several indicators and urban vitality beyond the sixth ring road of Beijing, China, including resident population density, road network density, and POI density [39]. Liang et al. found that POI density, road accessibility, building density, and house prices and salary levels had the most significant explanatory power on the urban vitality in Foshan, China [40]. Pan et al. found that the urban vitality in Macau, China, was mainly influenced by land use density, buildings, and public transportation [41]. Liu et al. found that the job–occupancy balance, floor area ratio, open space ratio, and road network density were positively associated with the urban vitality in Shanghai, China. Moreover, unlike most other studies, they argued that population density, land use mix, neighborhood size, rail station density, and accessibility were negatively related to urban vitality [42]. Some scholars have also performed comparative studies on cities with different development statuses. For example, Lu et al. compared the correlation between the urban vitality and built-up environment in Beijing and Chengdu, and found that POI density and public transportation accessibility were closely related to urban vitality, while density indicators had completely different effects on the vitality of cities with different development statuses [43]. In addition, some scholars have proposed a completeness index for complete streets as one of the key indicators of urban vitality, which consists of a set of indicators that help to assess the quality of free and public spaces in cities—especially on the streets, based on their planning functions: environment, place, and movement [44]. Xia et al. found that connectivity, compactness, building arrangement, iconic buildings, transport facilities, and open and green spaces had important effects on urban vitality via a comparative analysis in Chinese megacities, whereas land use mixture and building density presented limited or unintended effects. Furthermore, they suggested that some urban form indicators could contrarily contribute to the vitality for different cities, times, or dimensions [45]. In summary, many studies on the factors influencing urban vitality have been conducted. However, these studies have obtained different or even opposite results. Therefore, the influence of the built environment on urban vitality remains to be explored.

Early scholars usually used global regression models to perform influence factor analyses. The common global regression models include the Ordinary Least Squares Regression (OLS) model, Spatial lag regression (SLR) model, Spatial Error Regression (SEM) model, and spatial autoregressive model (SAR) model. For example, Zhu et al. [46] applied the OLS model and the model to investigate the factors influencing the urban vitality in Shenzhen, China. Long et al. [47] analyzed the effects of urban design on the economic vitality of 286 mega cities in China using the OLS model. Wu [22] and Xia [24] explored the factors influencing the urban vitality in different cities in China by applying the SLM model and SEM model, respectively. Indeed, global regression models can provide insights into the overall influencing factors on urban vitality, but ignore spatial heterogeneity. To address this limitation, Geographically Weighted Regression (GWR) models have been proposed to handle the spatial heterogeneity of relationships [48,49]. For example, Liu found that the spatial pattern and driving factors of the urban vitality in Nanjing, China, were spatially heterogeneous by using a GWR model [50]. Fan [51], Tu [52], and Zhang [18] compared the effectiveness of the OLS model and GWR model, and found that the latter achieved a better performance. However, it has been shown that the spatial distribution of urban vitality has a significant variation over 24 h within one day [53,54]. Compared to the GWR model, geographically and temporally weighted regression (GTWR) models have received attention from scholars, as they consider both temporal and spatial heterogeneity [53,55,56]. In recent years, some scholars have indicated that some built environment indicators may have a nonlinear effect on urban vitality, which means that the above-mentioned models cannot reveal the non-linear relationship between factors and urban vitality, and may lead to erroneous conclusions [21,34,53,57]. Scholars have typically investigated the non-linear relationship between them using machine learning regression algorithms such as random forests. However, machine learning algorithms have a black box nature and the explanation of the mechanism is not clear. Therefore, we applied the OLS model and the GTWR model to explore the complex spatio-temporal relationship between urban vitality

and built environment factors more precisely. Research models for influencing factors in previous studies are summarized in Table 2.

**Table 2.** Main research models for influencing factors.

| Literatures | Models | Key Factors |
|---|---|---|
| Sung et al., 2015 [11] | OLS | Land use mix, density, block size, building age, accessibility, and boundaries |
| Liu et al., 2022 [42] | OLS | Occupancy balance, floor area ratio, open space ratio, and road network density |
| Sulis et al., 2018 [36] | OLS | Population density and population mobility |
| Pan et al., 2021 [41] | SAR | Land use density, buildings, and public transportation |
| Wu et al., 2019 [22] | SLR | Mixed use and diversity, scale, older buildings, density, and boundary vacuum |
| Xia et al., 2022 [24] | SEM | Recreation POI density and transportation accessibility |
| Liu et al., 2020 [50] | GWR | Road accessibility, POI density, and POI diversity |
| Zhang et al., 2021 [56] | GTWR | POI density, POI diversity, road network density, and intersection density |

## 3. Materials and Methods

### 3.1. Study Area

Shenzhen was selected as the study area in this paper (see Figure 1). Shenzhen is a municipal city under the jurisdiction of Guangdong Province, located in the south of Guangdong Province, on the east coast of the Pearl River Delta, bordering the Hong Kong Special Administrative Region of China, Dongguan City, and Huizhou City. Shenzhen belongs to the north subtropical monsoon climate, full of sunshine and abundant rainfall. The city has a land area of 1997.47 km$^2$, with nine municipal districts, including Nanshan, Futian, Luohu, Yantian, Baoan, Guangming, Longhua, Longgang, and Pingshan, and one functional district, namely, Dapeng New District. Among them, Nanshan District, Futian District, and Luohu District are the administrative, scientific, technological, cultural, and financial centers of Shenzhen, with denser population distributions, while the other areas are mostly industrial land with sparser populations. Up to 2022, the resident population of Shenzhen was 17,681,600.

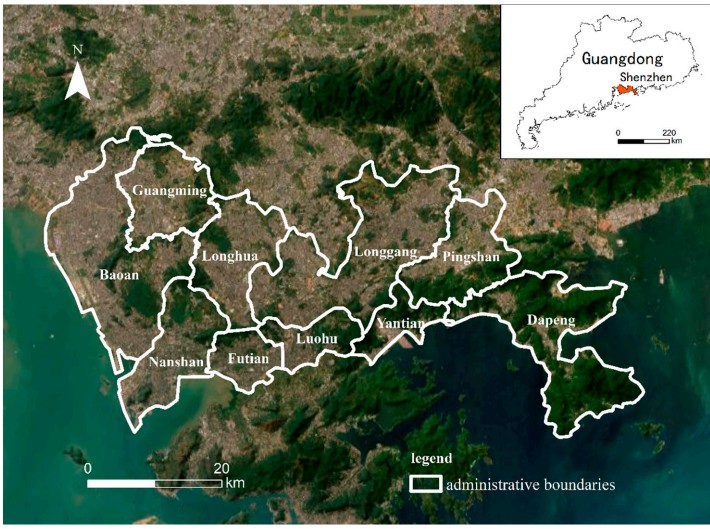

**Figure 1.** Study area.

### 3.2. Data Sources

The main data sources involved in our study are shown in Table 3.

**Table 3.** Data sources.

| Data | Source | Resolution (Format) | Time |
|---|---|---|---|
| Baidu heatmap | Baidu Map (https://huiyan.baidu.com/products/platform, accessed on 11 November 2020) | CSV | 2020 |
| Population density | WorldPop (https://www.worldpop.org/, accessed on 8 October 2022) | 1KM, Tiff | 2020 |
| POI | Amap (https://lbs.amap.com/, accessed on 12 September 2020) | CSV | 2020 |
| Roads | OpenStreetMap (https://www.openstreetmap.org/, accessed on 10 July 2020) | SHP | 2020 |
| Building boundaries | OpenStreetMap (https://www.openstreetmap.org/, accessed on 10 July 2020) | SHP | 2020 |
| Administrative district | Amap (https://lbs.amap.com/, accessed on 8 June 2020) | SHP | 2020 |

### 3.3. Traffic Analysis Zone

In this paper, we segmented the study area into irregulaulr polygons as Traffic Analysis Zone (TAZ) units based on the road network. Compared to the regular grid, TAZ units segmented from road networks are consistent with people's daily lives, thus making them more suitable for urban geography research [58,59]. The segmentation process of TAZ units based on road networks includes four steps. Firstly, the backbone network was established by selecting expressways, major roads, and minor roads, while eliminating fine inner-city branch roads. Secondly, fifty meter buffer zones of the remaining roads were created, and the intersecting areas were merged. Thirdly, the buffer zones were converted into a raster format and then simplified by extracting the centerline. Fourthly, line connections were processed, and suspension lines were removed to refine the analysis units according to the map. At last, 331 TAZ units were obtained for further analysis.

### 3.4. Urban Vitality Measurement Method

The raw Baidu user heatmap data are point elements and were not continuous over the study area. First, a kernel density analysis was performed to obtain vitality raster data. Then, zonal statistics were performed to obtain the mean vitality value for each TAZ unit. Finally, the statistical results were spatially linked with the TAZ units' vector data to obtain the vitality values of each TAZ unit in different time periods.

The standard deviation model is usually used to represent the degree of the dispersion of data [60,61]. In this paper, the standard deviation model was applied to analyze the degree of variation in urban vitality in each TAZ unit within one day. The standard deviation of vitality is calculated as follows:

$$S = \sqrt{\frac{\sum_{i=0}^{23}\left(V_i - \overline{V}\right)^2}{24}},\tag{1}$$

where $S$ represents the standard deviation of the vitality of a TAZ unit throughout the day, $V_i$ represents the vitality value of the TAZ unit at the $i$-th moment, and $\overline{V}$ represents the average vitality of the TAZ unit throughout the day.

### 3.5. The Urban Built Environmental Indicator System

The built environment is the man-made environment provided within a city to meet human activities and needs. We developed an urban built environmental indicator system based on the "5Ds" theory, which includes density, design, diversity, public transportation

accessibility, and destination accessibility [62]. The specific indicator system is shown in Table 4.

**Table 4.** Built environmental indicator system.

| Indicator | Variables | Description |
|---|---|---|
| Density | Population density (PD) | Mean population density within each TAZ unit |
| | Building density (BD) | Mean building density within each TAZ unit |
| Design | Road density (RD) | Road network density within each TAZ unit |
| Diversity | Urban functional mix (UM) | The level of diversity of POI types within each TAZ unit |
| Distance to transit | Bus stop density (SD) | Bus stop density within each TAZ unit |
| | Distance to metro (DM) | Distance of each TAZ unit from the nearest metro station |
| Destination accessibility | Distance to CBD (DC) | Distance of each TAZ unit from the nearest CBD |

### 3.6. The OLS Model

The OLS model is a classical linear regression model, which can effectively reveal the linear relationship between a dependent variable and independent variables. The principle of the OLS model is to continuously iterate to minimize the sum of the regression errors and thus obtain the optimal model. Its formula is:

$$y = \beta_0 + \sum_{i=1}^{n} \beta_i x_i + \varepsilon, \tag{2}$$

In Equation (2), $y$ denotes the dependent variable, $x_i$ denotes the independent variable, $\beta_i$ denotes the regression coefficient of the independent variable, $n$ denotes the number of independent variables, $\beta_0$ is a constant, and $\varepsilon$ is the error term.

### 3.7. The Moran's I Index

The presence of spatial autocorrelation (i.e., spatial non-stationary) of the model variables is a prerequisite for the application of a geographically weighted regression series model. The Moran's index (Moran's I) can test the spatial autocorrelation of variables [63], with a value between −1 and 1. The closer the value is to 1 or −1, the higher the positive or negative spatial autocorrelation of the variable. A value of 0 indicates that the variables are randomly distributed in space. The formula of Moran's I index is as follows:

$$I = \frac{n}{S_0} \frac{\sum_{i=1}^{n} \sum_{j=1}^{n} w_{i,j} z_i z_j}{\sum_{i=1}^{n} z_i^2}, \tag{3}$$

In Equation (3), $z_i(z_j)$ denotes the deviation of the value of element $i$ ($j$) from its mean, $w_{i,j}$ is the spatial weight between element $i$ and element $j$, $n$ denotes the total number of samples, and $S_0$ is the aggregation of the spatial weights.

### 3.8. The GTWR Model

The OLS model only estimates the coefficients of the variables on a global scale [64], which cannot reveal the spatial and temporal variation in the coefficients. Compared to the traditional GWR model, the GTWR model is more suitable for a regression analysis of spatio-temporal data by considering both temporal and spatial heterogeneity. The GTWR model was proposed by Fotheringham in 2015, which reflects the differences in the impacts of independent variables on dependent variables in time and space simultaneously by establishing different regression coefficients for different times and spaces [65]. Therefore, the GTWR model can deal with the non-stationarity of variables in time and space to some extent. The equation of the GTWR model is as follows:

$$y_i = \beta_0(u_i, v_i, t_i) + \sum_{k=1}^{d} \beta_k(u_i, v_i, t_i) x_{ik} + \varepsilon_i \tag{4}$$

In Equation (4), $y_i$ denotes the dependent variable, $(u_i, v_i, t_i)$ denotes the spatio-temporal coordinates of the sample $i$, $\beta_0(u_i, v_i, t_i)$ denotes the intercept, $\beta_k(u_i, v_i, t_i)$ denotes the regression coefficient of variable $k$ in sample $i$, $x_{ik}$ denotes the observed value of variable $k$ in sample $i$, and $d$ denotes the number of variables.

## 4. Results

### 4.1. Spatio-Temporal Variation of Urban Vitality

In our study, eight representative moments of urban vitality, including 0:00 a.m., 3:00 a.m., 6:00 a.m., 9:00 a.m., 12:00 a.m., 15:00 p.m., 18:00 p.m., and 21:00 p.m., were selected for the analysis. We believed that these moments cover the main time points of the daily lives of residents. The visualization results of vitality on weekdays and weekends are shown in Figures 2 and 3, respectively.

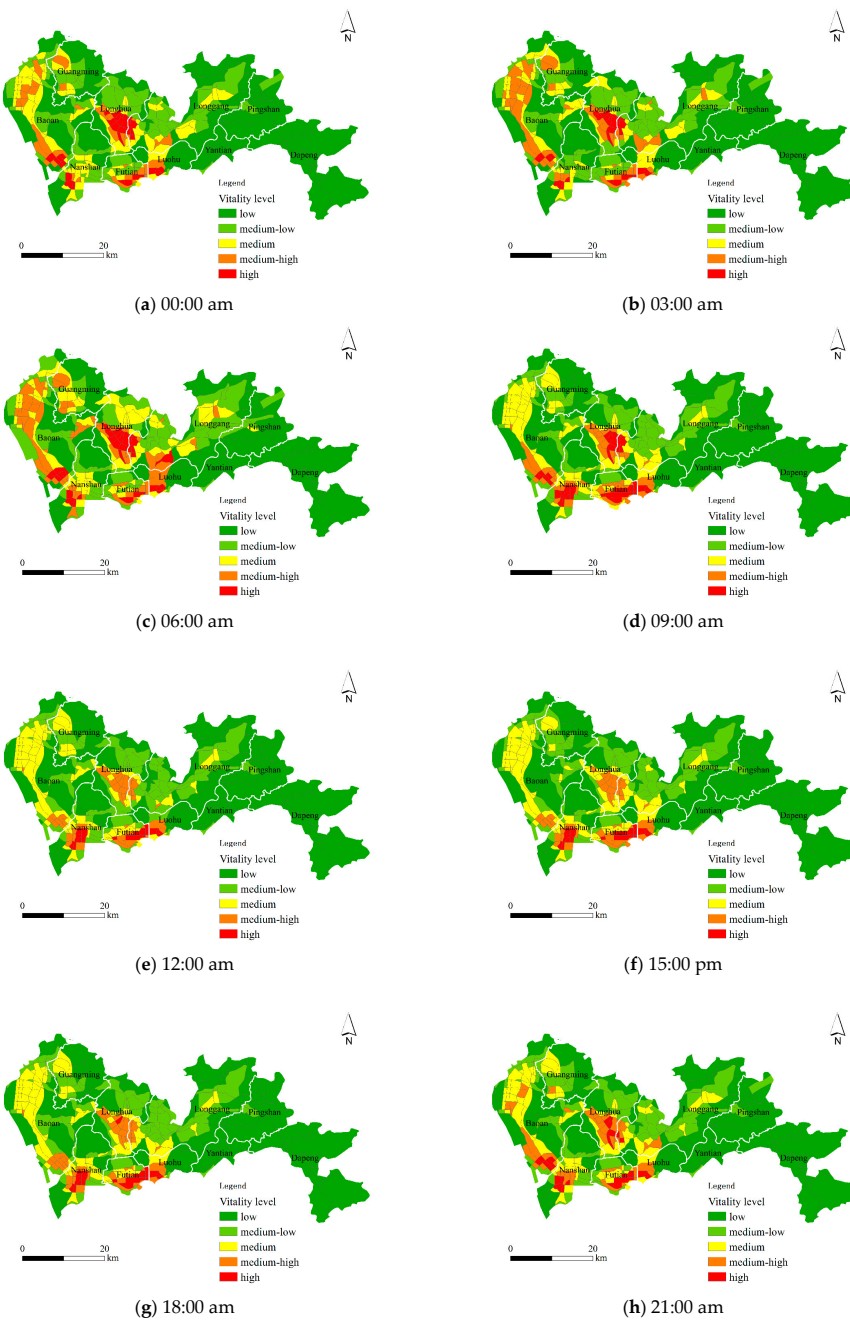

**Figure 2.** The dynamic of urban vitality in Shenzhen on weekday.

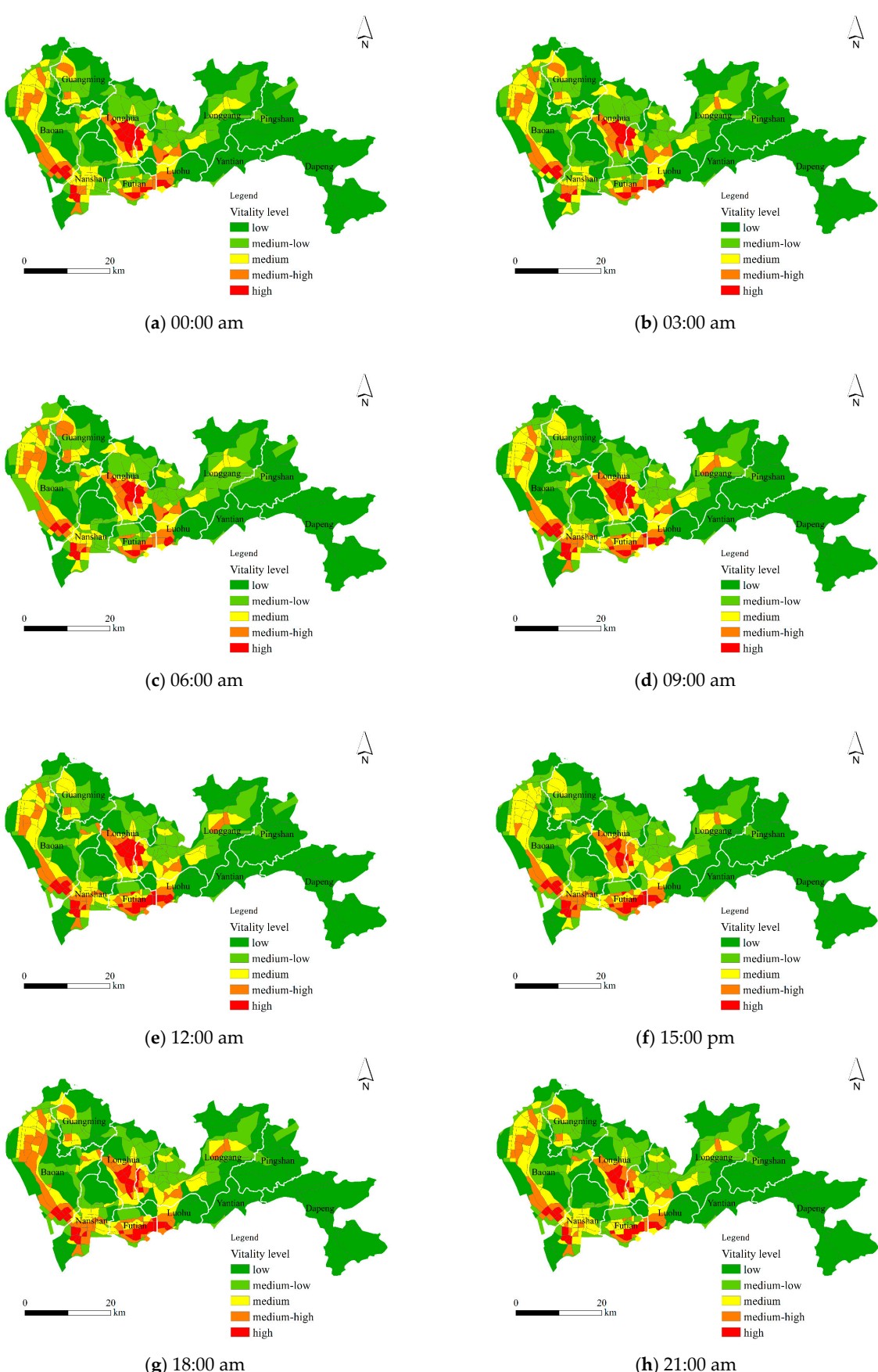

**Figure 3.** The dynamic of urban vitality in Shenzhen on weekend.

The change in urban vitality during the weekday can be divided into the following periods. The first period is from 0:00 a.m. to 6:00 a.m., with the residents mainly living in residential areas. It can be observed that there is a noticeable trend of decreasing vitality levels in the southern part of Shenzhen, which gradually transitions into lower-vitality-level areas towards the periphery. However, the Longhua District stands out as having the main concentration of high-vitality-level areas during this period. Furthermore, improvements in vitality levels were observed in the northern part of Baoan District and Dafen Street in Longgang District. On the other hand, the vitality level of the northern part of Longhua District decreased. In short, there was a clear trend of urban vitality shifting from the south to the north, indicating an obvious separation between areas of occupation and residences in Shenzhen.

The second period is from 6:00 a.m. to 12:00 p.m., when residents primarily commute from their residential areas to their workplaces. During this period, the urban vitality level in the northern part of Baoan District, the northern part of Longhua District, Longsheng Street, and Wuhe Street decreased. High-grade vitality units shifted from north to south and were concentrated in Nanshan District, Futian District, and Luohu District. The distribution of urban vitality was more concentrated compared to the distribution in the night time period.

The third period is from 12:00 p.m. to 18:00 p.m., in which citizens are primarily engaged in activities at their workplaces. Compared to the previous period, the distribution of urban vitality did not exhibit significant changes. During this period, the high-grade vitality units were concentrated in Nanshan District, Futian District, and Luohu District. Additionally, the secondary high-grade vitality units were distributed in the Bao'an Central Area, Longsheng Street, and Wuhe Street.

The fourth period is from 18:00 p.m. to 21:00 p.m., when people usually leave their workplace and return to their residence. The distribution pattern of urban vitality shifted from centralized to decentralized, and the hotspot areas moved to the surrounding areas. Specifically, urban vitality decreased in areas such as the surroundings of Shenzhen University and the old streets of Luohu District, while it increased in areas such as the Qianhai Free Trade Zone, Bao'an Central District, Buji Street, and Dafen Street.

Compared to weekdays, residents' activities were more random on weekends due to the absence of commuting, which, in turn, leads to a more spatially dispersed distribution of vitality. Unlike weekdays, the characteristic of stage division throughout the day was not prominent during weekends. During the period from 0:00 a.m. to 9:00 a.m., the distribution of urban vitality did not change significantly, and the high-vitality areas were mainly concentrated in Longsheng Street, Wuhe Street, Xixiang Street, and Bao'an Central District. In addition, there were also a small number of high-grade vitality units located in Futian District, Nanshan District, and Luohu District. The distribution of secondary-grade vitality units was more dispersed. Compared to the previous period, the urban vitality showed a significant southward trend between 12:00 p.m. and 15:00 p.m., with a noticeable increase in high vitality levels in Futian District. As one of the most prosperous areas in Shenzhen, Futian District has many commercial and entertainment facilities, which are more attractive to residents, and thus it has become the main activity place for residents on weekends. In addition, there were a small number of TAZ units with a secondary high vitality level located in the surrounding of Longcheng Square in Longgang District, indicating that residents prefer to visit these areas farther away from the central area for leisure activities on weekends. Between 18:00 p.m. and 21:00 p.m., the distribution of urban vitality in Shenzhen changed again from a concentrated pattern to a dispersed pattern. Overall, the urban vitality in Shenzhen was more dispersed in terms of spatial distribution during weekends, and there was no obvious temporal pattern change. This result further indicated that citizens' travel activities on weekends were more flexible and diverse.

The Standard deviation distribution of vitality on weekdays and weekends is shown in Figure 4.

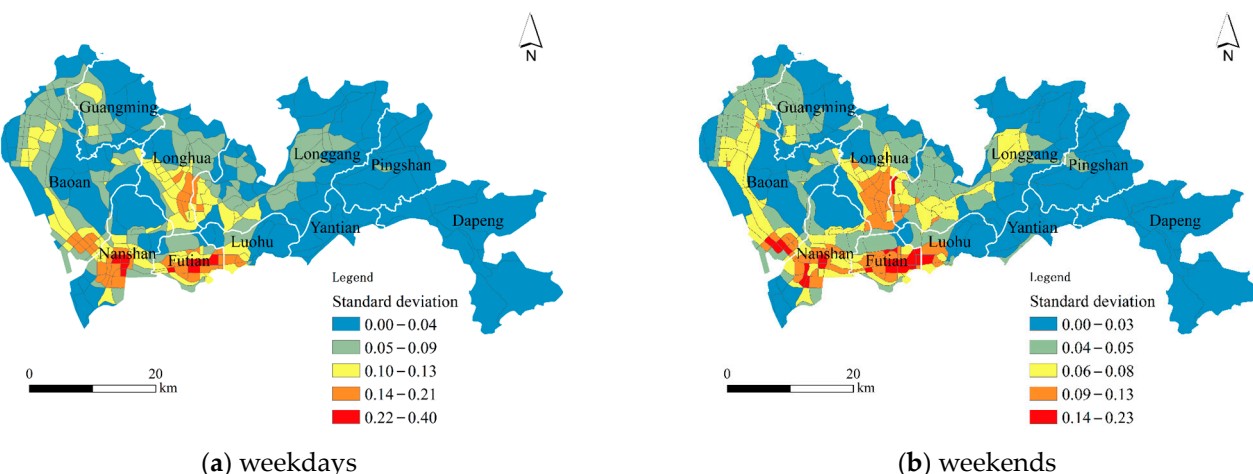

(**a**) weekdays                                                                                                    (**b**) weekends

**Figure 4.** Standard deviation distribution of vitality on weekdays and weekends.

As can be seen From Figure 4 that the standard deviation of weekdays was significantly larger than that of weekends, indicating higher variations in vitality throughout the weekdays. The standard deviation distributions of weekdays and weekends were similar, and both were positively correlated with the urban vitality distribution overall. That is, the TAZ units with a high standard deviation of vitality also exhibited high vitality during weekdays and weekends. On weekdays, TAZ units with a high standard deviation of vitality were concentrated around the Shenzhen University (which is in Nanshan District) and the Convention and Exhibition Center (which is in Futian District), indicating that the vitality of these areas changed more drastically on weekdays. In addition, there were a few TAZ units with a high standard deviation in Baoan Center in Baoan District, Old Street in Luohu District, and Longsheng Street in Longhua District, indicating noticeable vitality fluctuations in these areas. Compared to weekdays, the TAZ units with a relatively high standard deviation of vitality on weekends were more widely distributed. Besides Nanshan District and Futian District, Luohu District, Baoan District, and Longgang District also had a small number of TAZ units with a relatively high standard deviation of vitality. Moreover, the vitality of areas such as Longsheng Street, Wuhe Street, and the vicinity of Shenzhen North Railway Station also changed relatively drastically during weekends. The urban vitality of other areas did not change significantly throughout weekends.

*4.2. Global Regression Results of the Association between Built Environment and Urban Vitality*

The application of the OLS model on a global scale is the first step of a spatial regression analysis. As we know, multicollinearity can significantly affect the rationality of the regression model. Therefore, the independent variables should be tested for multicollinearity at first. An OLS model was established with urban vitality data as the dependent variable and built environment indicators as independent variables, after standardization. The problem of multicollinearity among the variables was examined by checking the tolerance and variance inflation factor (VIF) of each factor. When the value of VIF does not exceed 10 and the tolerance is less than 0.1, it can be assumed that there is no serious multicollinearity problem in the model variables. The results of the multicollinearity diagnosis showed that the tolerances of all the variables were greater than 0.1 and the VIF values were less than 10. Therefore, we thought that the data met the requirements of the OLS model.

The OLS regression models for both weekdays and weekends are illustrated in Figure 5. The adjusted R-squared (adj. $R^2$) values of each model were approximately 0.7, indicating a good fitness of the models. The coefficient values of the model showed that the built environment factors had different impacts on urban vitality, and the impacts varied over time. During weekdays, population density, building density, and bus stop density showed significant positive correlations with urban vitality, indicating that these factors had a consistent enhancing effect on urban vitality during weekdays. Among them, the building

density showed a higher effect during nighttime than daytime. The distance to the CBD had a significant negative relationship with urban vitality throughout weekdays. This result implied that proximity to the CBD leads to a higher urban vitality and vice versa. Therefore, it can be inferenced that the CBD plays a vital role in enhancing urban vitality, whether during daytime or nighttime. Furthermore, road network density and functional mix showed positive and negative effects, respectively, only during the daytime. The distance to the metro station had a significant positive effect on urban vitality, except for the period from 2:00 a.m. to 7:00 a.m.

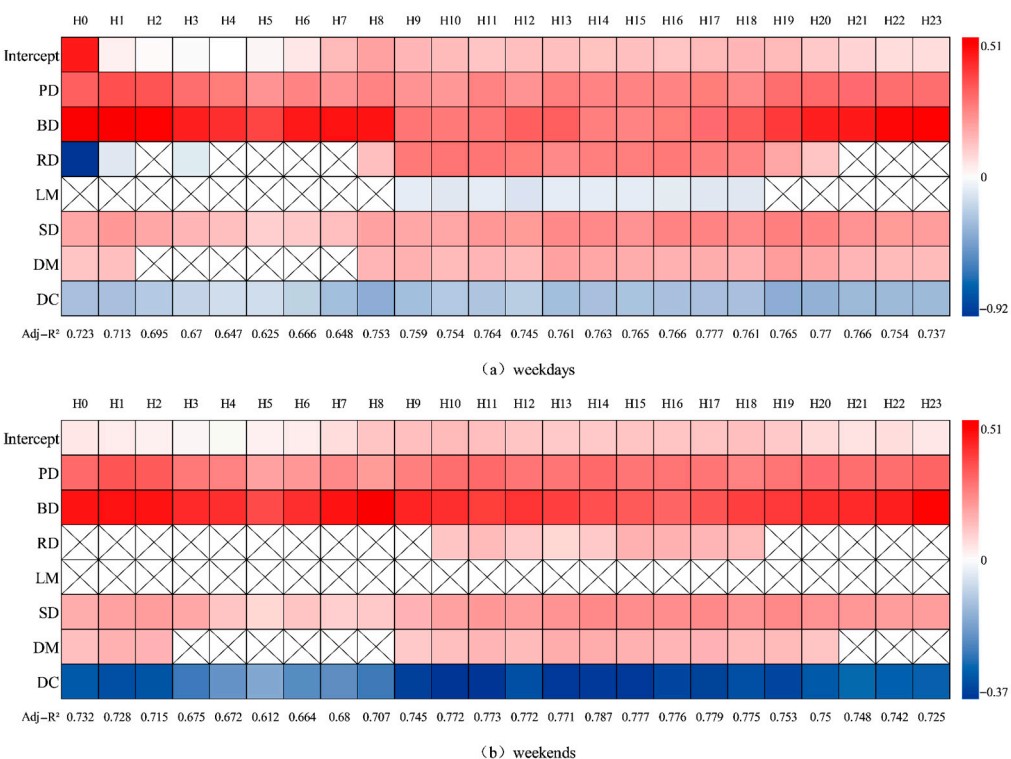

**Figure 5.** Regression results of OLS model for different moments on weekdays and weekends.

Like during weekdays, population density, building density, and bus stop density also showed significant positive correlations with urban vitality throughout the weekends, indicating that these factors also have an enhancing effect on urban vitality. Distance to the CBD also showed a significant negative correlation over the weekends. Unlike weekdays, road network density showed a significant positive correlation only from 10:00 a.m. to 18:00 p.m. The effect of distance to metro station on urban vitality showed a positive correlation from 0:00 to 2:00 a.m. and from 9:00 a.m. to 20:00 p.m. There was no significant correlation between functional mix and urban vitality throughout the weekends.

In summary, there were large differences in the effects of built environment factors on urban vitality, and some variations over time. Therefore, we suggested that the temporally non-stationarity of the effects of built environment factors on urban vitality should not be ignored.

### 4.3. Spatio-Temporal Geographical Regression Results of the Association between Built Environment and Urban Vitality

#### 4.3.1. Spatial Autocorrelation Test

The presence of spatial non-stationarity in model variables is a prerequisite for the usage of geographically weighted regression techniques. In our study, we applied the Moran's I index to test whether each built environment variable was spatially autocorrelated. Moran's I has a value between −1 and 1. A value closer to 1 indicates a higher positive spatial autocorrelation of the variable. Conversely, the higher the negative auto-

correlation of the variable. A value of 0 indicates that the variables are almost randomly distributed in a study area. The results of the spatial autocorrelation test are shown in Tables 5 and 6.

**Table 5.** Hourly spatial autocorrelation test results for weekdays.

| Time | H0 | H1 | H2 | H3 | H4 | H5 | H6 | H7 | H8 | H9 | H10 | H11 |
|---|---|---|---|---|---|---|---|---|---|---|---|---|
| Moran's I | 0.36 | 0.36 | 0.35 | 0.35 | 0.34 | 0.33 | 0.34 | 0.40 | 0.45 | 0.54 | 0.52 | 0.51 |
| *p*-value | 0.00 | 0.00 | 0.00 | 0.00 | 0.00 | 0.00 | 0.00 | 0.00 | 0.00 | 0.00 | 0.00 | 0.00 |
| **Time** | **H12** | **H13** | **H14** | **H15** | **H16** | **H17** | **H18** | **H19** | **H20** | **H21** | **H22** | **H23** |
| Moran's I | 0.46 | 0.49 | 0.52 | 0.53 | 0.52 | 0.50 | 0.48 | 0.48 | 0.45 | 0.42 | 0.39 | 0.38 |
| *p*-value | 0.00 | 0.00 | 0.00 | 0.00 | 0.00 | 0.00 | 0.00 | 0.00 | 0.00 | 0.00 | 0.00 | 0.00 |

**Table 6.** Hourly spatial autocorrelation test results for weekends.

| Time | H0 | H1 | H2 | H3 | H4 | H5 | H6 | H7 | H8 | H9 | H10 | H11 |
|---|---|---|---|---|---|---|---|---|---|---|---|---|
| Moran's I | 0.38 | 0.38 | 0.37 | 0.36 | 0.35 | 0.34 | 0.35 | 0.36 | 0.36 | 0.43 | 0.45 | 0.47 |
| *p*-value | 0.00 | 0.00 | 0.00 | 0.00 | 0.00 | 0.00 | 0.00 | 0.00 | 0.00 | 0.00 | 0.00 | 0.00 |
| **Time** | **H12** | **H13** | **H14** | **H15** | **H16** | **H17** | **H18** | **H19** | **H20** | **H21** | **H22** | **H23** |
| Moran's I | 0.44 | 0.45 | 0.46 | 0.49 | 0.49 | 0.47 | 0.44 | 0.43 | 0.40 | 0.39 | 0.38 | 0.36 |
| *p*-value | 0.00 | 0.00 | 0.00 | 0.00 | 0.00 | 0.00 | 0.00 | 0.00 | 0.00 | 0.00 | 0.00 | 0.00 |

From Tables 5 and 6, we can see that the Moran's I values of urban vitality were greater than 0.3 on both weekdays and weekends, and the *p*-values were less than 0.01. This result indicated that there was a significant positive spatial autocorrelation in the distribution of urban vitality on both weekdays and weekends in Shenzhen. Therefore, a spatio-temporal geographically weighted regression model is needed to address the issue of spatio-temporal non-stationarity.

### 4.3.2. Results of GTWR Regression Analysis

GTWR models were developed for weekdays and weekends, respectively. The statistical results of the regression coefficients of each variable at different hours are shown in Table 7.

**Table 7.** GTWR regression coefficient statistics.

| Indicators | Weekdays | | | | Weekends | | | |
|---|---|---|---|---|---|---|---|---|
| | Min | Max | Average | Std. | Min | Max | Average | Std. |
| PD | 0.133 | 0.967 | 0.283 | 0.137 | 0.133 | 0.967 | 0.283 | 0.137 |
| BD | 0.053 | 0.698 | 0.385 | 0.142 | 0.053 | 0.698 | 0.385 | 0.142 |
| RD | −0.367 | 0.306 | 0.09 | 0.131 | −0.367 | 0.306 | 0.09 | 0.131 |
| MIX | −0.217 | 0.262 | 0.029 | 0.094 | −0.217 | 0.262 | 0.029 | 0.094 |
| SD | −0.186 | −0.432 | 0.169 | 0.082 | −0.186 | −0.432 | 0.169 | 0.082 |
| DM | −0.159 | 0.513 | 0.066 | 0.125 | −0.159 | 0.513 | 0.066 | 0.125 |
| DC | −0.481 | 0.249 | −0.221 | 0.121 | −0.481 | 0.249 | −0.221 | 0.121 |

The Adj.$R^2$ of two GTWR models during weekdays and weekends were 0.811 and 0.801, respectively, indicating that the GTWR models could explain more than 80% of the observed sample information, which is significantly better than the OLS model. Unlike the OLS model, the coefficients of the independent variables in the GTWR model change according to the spatial and temporal coordinates of the sample. Therefore, the relationship between built environment factors and urban vitality can be interpreted more precisely from both temporal and spatial perspectives.

### 4.3.3. Spatial–Temporal Analysis of GTWR Regression Coefficients

- Temporal analysis of GTWR regression coefficients

To better understand the spatiotemporal heterogeneity of the relationship between built environment factors and urban vitality, the 24 h averages of coefficients and the averages of 331 TAZ units were calculated for weekdays and weekends, respectively. A visualization of the statistical results is shown in Figure 6.

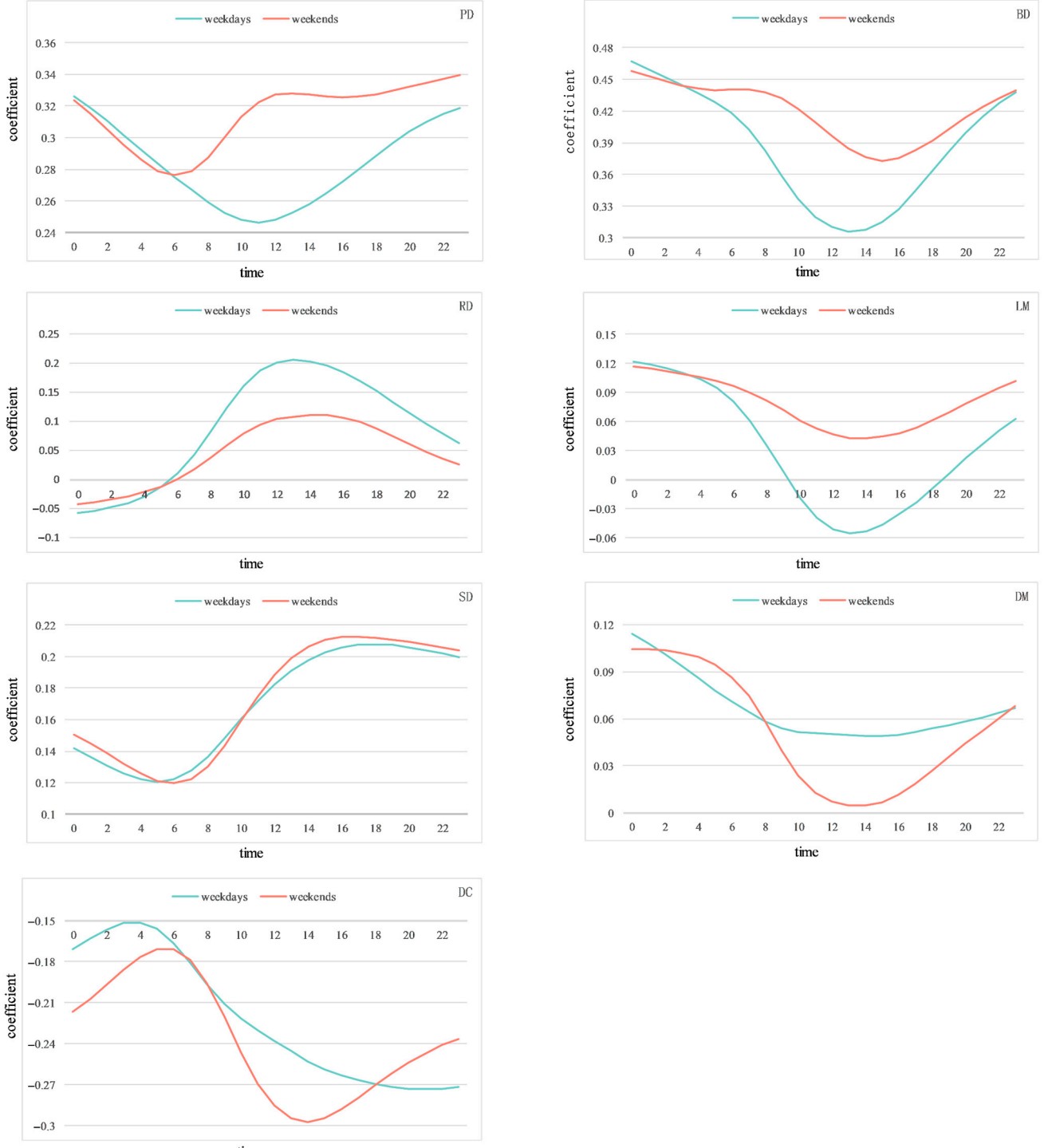

**Figure 6.** Regression coefficients for different moments on weekdays and weekends.

From an overall perspective, the regression coefficients of population density and building density were positive on weekdays and weekends, indicating a positive effect on urban vitality. Population is the main body of the urban system, and buildings, as one of the main places of human activities, can reflect the development of a city to a certain extent. Therefore, population density and building density usually have a significant positive effect on urban vitality. The coefficients of population density and building density showed a trend of first decreasing and then increasing throughout the day. On weekdays, the absolute value of the coefficient of population density reached its lowest value at 11:00 a.m., while the absolute value of the coefficient of building density reached its lowest value at 13:00 p.m. This indicated that the effects of population density and building density on urban vitality were enhanced in the afternoon. On weekends, the difference in the change characteristics between the coefficients of population density and building density was significant. Throughout the weekends, the coefficient of population density exhibited a characteristic pattern of initially decreasing, then increasing, and finally stabilizing. Among them, the influence of population density on urban vitality gradually decreased during the nighttime. After 6:00 a.m., the influence of population density gradually increased and finally reached its maximum at around 12:00 p.m. and stabilized thereafter.

The absolute values of the coefficient of building density maintained a high value until 6:00 a.m., and then showed a trend of initially decreasing and then increasing, indicating that building density showed a stronger effect during nighttime than daytime. It was noticed that the absolute values of the coefficients of population density and building density were generally higher on weekends than on weekdays. One possible explanation is that, compared to weekdays, on weekends, residents tend to have a higher proportion of leisure and recreational activities usually happening in commercial areas with a high building density. Thus, the density indicator had a stronger effect on them.

On weekdays and weekends, the coefficients of road network density were positive, except for the period from 0:00 a.m. to 6:00 a.m., indicating that there was a significant diurnal difference in the influence of road network density on urban vitality. That is, road network density showed a significant positive correlation with urban vitality during the daytime and peaked at around 14:00 p.m. The possible reason for this is that people are still mainly active during the day. The absolute values of the road network density coefficients during weekdays were greater than those of weekends, indicating that road network density had a stronger effect on urban vitality during weekdays. This finding can be explained by the fact that residents need to commute to their workplaces and engage in various activities on weekdays. Residents have more freedom in their travel arrangements on the weekends, so the effect of road network density on urban vitality was slightly weaker.

The absolute value of the coefficient of functional mix was consistently low, indicating that functional mix had little impact on urban vitality. Furthermore, there was a notable difference in the coefficient values of functional mix between weekdays and weekends. On weekdays, the functional mix coefficient was positive, except for the period from 10:00 a.m. to 18:00 p.m. It is possible that the effect of functional mix was not reflected, since the period between 10:00 a.m. and 18:00 p.m. is usually working hours, which is dominated by the work activities of urban residents. However, functional mix had a positive effect on urban vitality during weekends. It can be inferenced that residents preferred to visit commercial centers or scenic areas on weekends, which usually have a high degree of mix.

The coefficients of bus stop density were positive on weekdays and weekends, and the variation characteristics were very similar, suggesting that bus stop density had a significant effect on urban vitality. Bus stop density had a stronger effect on urban vitality during the daytime than the nighttime. A possible explanation is that, as the public transport mode with the widest coverage, buses are the preferred transport mode for many residents and thus have a significant enhancement effect on urban vitality. Since most bus routes stopped running at night, the intensity of their effect was higher during the daytime than the nighttime.

Unlike bus station density, distance to metro stations had a positive correlation with urban vitality. That is, the further the distance from the metro station, the higher the urban vitality. This finding challenges our perception that proximity to metro stations would enhance urban vitality. A possible explanation is that the metro station is often not the ultimate destination, and residents usually need a feeder trip after getting off the metro. The value of the coefficient on the weekends showed a trend of initially decreasing and then increasing, reaching the lowest value at 14:00 p.m.

The coefficient of distance to the CBD was negative on weekdays and weekends, indicating that distance to the CBD consistently had negative impact on urban vitality. On weekdays, the absolute value of the coefficient stayed at a low value between 0:00 a.m. and 6:00 a.m., and increased slowly after 6:00 a.m., indicating that the CBD had a stronger influence on urban vitality during the daytime than the nighttime. The reason may be that Shenzhen's CBD is a comprehensive area for technology, finance, politics, and other functions. During the daytime on weekdays, residents usually gather in the CBD for working and living, so the closer the area to the CBD, the higher the urban vitality. On weekends, the absolute value of the coefficient was also high during the daytime and low during the nighttime, indicating that the CBD is also attractive to residents on weekends and has a boosting effect on urban vitality.

- Spatial analysis of GTWR regression coefficients

The 24 h averages of the model coefficients for each TAZ unit on weekdays and weekends were calculated, and the visualization results are shown from Figures 7–13. The regions with positive regression coefficients are represented by warm tones and vice versa by cool tones. The impact of built environment factors on urban vitality showed significant differences with the TAZ units. The comparison showed that the distribution patterns of the same built environment factor on weekdays and weekends were roughly similar, indicating that the differences in the effects of individual built environment factors depended more on the differences between TAZ units.

(1)  Density

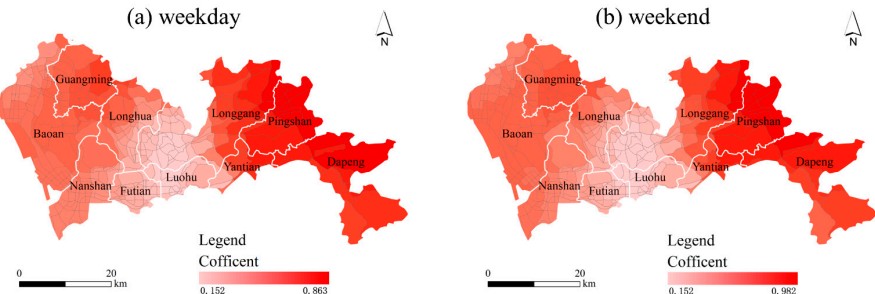

**Figure 7.** Coefficient map of population density.

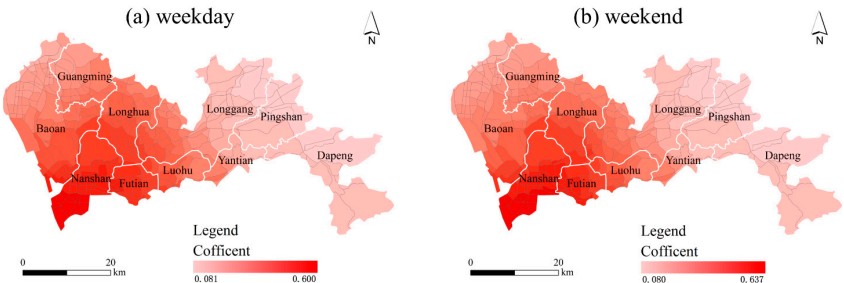

**Figure 8.** Coefficient map of building density.

The effect of population density on urban vitality was strongest in the eastern part of Shenzhen, including the Pingshan District, Dapeng New District, Longgang District, and

the eastern part of the Yantian District. These areas are suburban regions with a relatively lower population density. Therefore, increasing the distribution of population in these areas is beneficial for the improvement of urban vitality. In the central and southern parts of Shenzhen, the intensity of the effect of population density was lower, including the central areas of Futian District, Nanshan District, and Luohu District. These areas are the core areas of Shenzhen, where industries and businesses are more developed and have a strong attraction to urban residents. Therefore, increasing the population density in central areas may have some effect on urban vitality, but it is not as significant as that in suburban regions.

Unlike population density, the effect of building density showed a clear circular structure, with coefficient values decreasing in all directions centered on the Shenzhen CBD. This finding suggested that proximity to the CBD played a crucial role in the relationship between building density and urban vitality. The reason may be that the CBD of Shenzhen is a vibrant area that encompasses various sectors such as technology, finance, and entertainment, making it highly attractive to residents. This area is more densely populated but has a relatively less built-up area, so an appropriate increase in buildings will have a significant effect on urban vitality. In contrast, the effect of increasing building density on the urban vitality in non-core areas of Shenzhen was less pronounced.

(2) Design

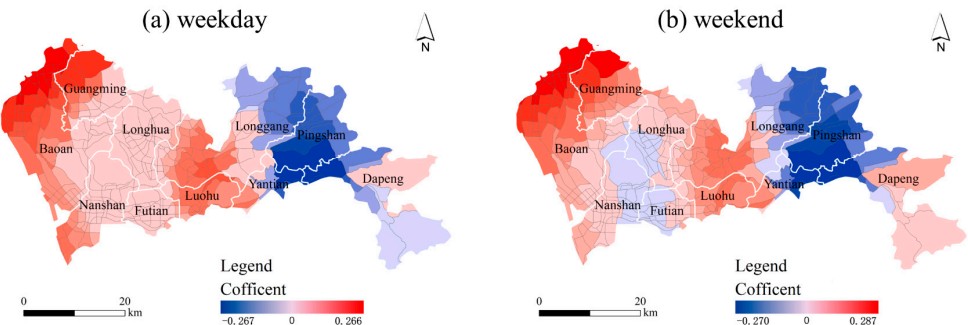

**Figure 9.** Coefficient map of road density.

The effect of road network density on the urban vitality in Shenzhen exhibited spatial variations, and there were subtle differences between the patterns on weekdays and weekends. On weekdays, the effect of road network density on urban vitality showed a positive effect in the central and western parts of the city, and it was especially strongest in the northwest. Thus, increasing the road network density in these areas can promote more population activities to increase urban vitality. In the eastern parts of the study area, road network density showed a negative effect. This may be because the east of Shenzhen is dominated by mountains, forests, and green areas with fewer populations. On the weekends, the negative impact areas besides the eastern part also contained Nanshan District, Futian District, and part of Longhua District. The reasons could be related to reduced commuting needs and a shift in residents' activities towards leisure and recreational areas, where road network density may have a relatively smaller influence.

(3) Diversity

Functional mix exhibited both positive and negative effects on urban vitality on weekdays and weekends. The positive impact areas were mainly concentrated in the eastern and western parts of Shenzhen, especially in four districts such as Bao'an District, Guangming District, Nanshan District, and Longhua District. Therefore, appropriately increasing the degree of mixed land use in such areas can promote urban vitality. On the other hand, the negative impact areas were mainly concentrated in the central and western parts of Shenzhen, including Luohu District, Yantian District, Longgang District, and Pingshan District. Among them, the negative effect was most prominent in the eastern parts of the Yantian and Luohu districts. The possible explanation is that such areas

not only have a certain distribution of industrial and commercial facilities, but also have many green areas such as mountains and parks, and thus have more diverse land use types. The negative effect of functional mix on urban vitality in these regions may indicate that the effects of some factors, such as limited accessibility or inadequate infrastructure, outweighed the positive influence of mixed land use.

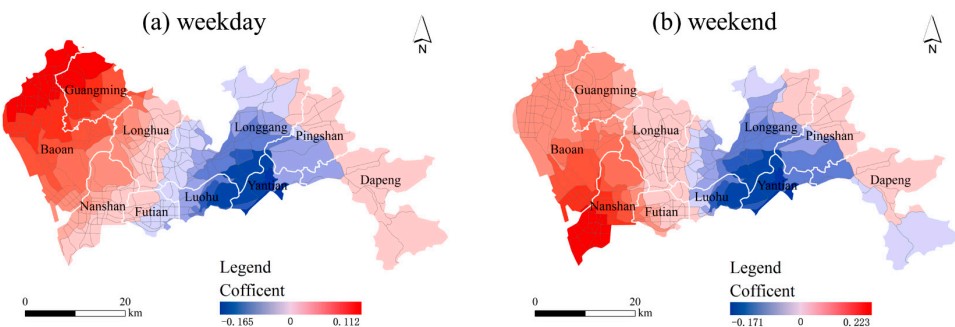

**Figure 10.** Coefficient map of functional mixture.

(4)     Distance to transit

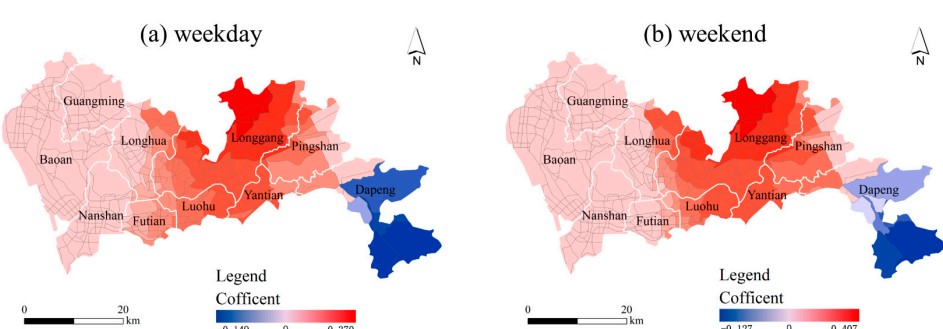

**Figure 11.** Coefficient map of bus stop density.

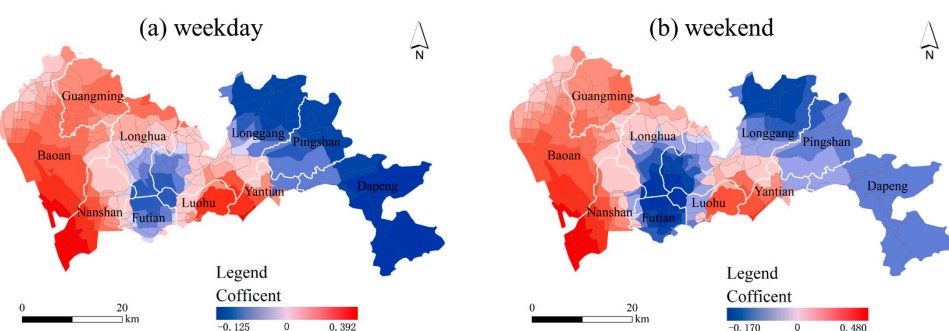

**Figure 12.** Coefficient map of distance to metro.

Except for Dapeng new district, the effects of bus stop density on the urban vitality in the study area were positive. Therefore, increasing the deployment of bus stops in these areas can contribute to urban vitality. It is worth noting that the effect of bus stop density on urban vitality was most significant in Yantian district and Longgang district. The possible explanation is that Yantian district and Longgang district have fewer subway stations compared to the western part of Shenzhen; thus, buses have become the preferred transportation mode for residents. Therefore, the effect of increasing the density of bus stops in such areas would be more prominent. The variation in the coefficient values of distance from metro stations was more pronounced within the study area than the density of bus stops. In the eastern and central parts of Shenzhen, the coefficient of distance to the metro station was negative, indicating that, the closer to the metro station in such areas, the higher the urban vitality. These areas included suburban areas such as Pingshan District,

Dapeng New District, and eastern Longgang District, as well as more developed areas such as Futian District and Longhua District. This result indicated that the opening of new subway stations had a certain promotion effect on the vitality in these areas. Outside of these areas, the coefficient of distance to the metro station was positive, especially in the southwest. This result implied that the accessibility of metro stations in such areas may not have a promoting effect on urban vitality.

(5)     Destination accessibility

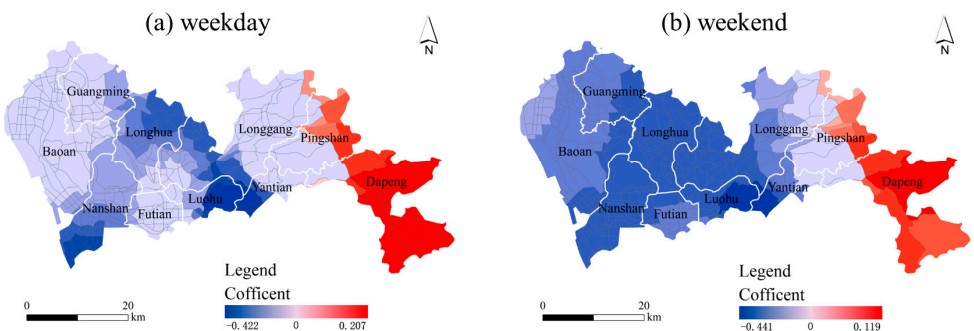

**Figure 13.** Coefficient map of distance to CBD.

Except for in Dapeng New District and the eastern part of Pingshan District, distance from the CBD mainly showed a negative correlation with urban vitality. This means that, the closer an area was to the CBD, the higher its urban vitality. The CBD serves as the hub for economic, technological, and cultural activities, and benefits from excellent municipal transportation and infrastructure. As a result, it plays a crucial role in enhancing urban vitality. However, it is important to note that the influence of the CBD on surrounding areas is somewhat limited. To this end, we proposed building a polycentric urban structure within Shenzhen to improve the accessibility of the entire region to the CBD. By establishing multiple centers of economic, social, and cultural activities in different parts of the city, the vitality of the total area can be enhanced.

The above results revealed that urban vitality was the result of the complex effects of different built environment factors, and different regions were affected due to their own characteristics. Therefore, urban planning strategies should be developed according to the actual situation of each region to enhance urban vitality.

## 5. Discussion

### 5.1. Research Novelty and Contribution

Our findings can prompt the understanding of the complex relationship between urban vitality and the built environment in the prospect of spatiotemporal heterogeneity. Compared to other studies in the same area, this paper has some novelty in the analysis scale of TAZ units and the application of a GTWR model, leading to novel conclusions that have not been discussed in previous studies. For example, Ouyang et al. found that a significant positive correlation exists between the stability and intensity of vitality in Shenzhen, and that high-stability areas were mainly distributed at the center of city [66]. However, they did not explore the influence of the built environment on urban vitality. Lv et al. analyzed the block vitality in Shenzhen by using multi-source big data and a GWR model and found that distance to subway station, road density, residential density, land mixed use, and compactness had significant influences on block vitality, but the influence varied from block to block, showing a strong spatial heterogeneity [67]. Zhang et al. assessed the vitality of Shenzhen in the scale of a grid and found that dense road networks, abundant transportation, and commercial, recreational, entertainment, sports, and leisure facilities were positive indicators of vitality in Shenzhen, while urban villages and residential areas were negative indicators [68]. The spatial resolutions of their study were finer than those of our study. However, they neglected temporal heterogeneity in the

effect of the built environment on urban vitality. Thus, our paper obtained some results that differ those of from previous studies. In our study, we found that road density and mixed land use had both positive and negative influences on urban vitality, while Lv et al. and Zhang et al. only found positive influence. The negative influence area found in our study was mainly the northeastern area of Shenzhen that is far away from the center of city. The reason may be that the northeastern part of Shenzhen is less developed, and residents rely more on public transportation, such as the subway, instead of cars for commuting. Therefore, a denser road network did not attract more people from other areas. Wu et al. found that urban function, accessibility, building form, and human perceptions were the main factors that influence urban vitality, and the influences during daytime and nighttime exhibited obvious differences [69]. This paper distinguished a 24 h characterization of the effect of the built environment indicators on vitality on weekdays and weekends, and the results were more refined compared to Wu et al.'s study. Moreover, the data for measuring vitality in Wu et al.'s study were the Tencent EasyGo data collected in 2019, which do not reflect the latest characteristics of citizens' activities. It is worth noting that the analysis unit in previous studies was dominated by the grid, which is not consistent with the boundaries of residents' activities in real life. Lastly, we suggested that, to optimize the urban layout for improving vitality, local authorities and planners should understand the complex relationship between the built environment and urban vitality at fine spatiotemporal scales based on multi-source "human sensing" data.

### 5.2. Urban Planning Implications

The planning insights based on the analysis of urban vitality are as follows.

The first recommendation is to build a "high-density" but "not crowded" city. According to the effect of population density on urban vitality, it is suggested that the population density of Pingshan District and Dapeng New District can be appropriately increased to improve the total vitality of the city. According to the effect of building density, it is suggested to increase the density of building facilities in the core area of Shenzhen to enhance urban vitality more efficiently. The suggestion can provide a reference for local authorities and real estate developers to make land development plans.

The second recommendation is to optimize the road network layout in the city. According to the spatiotemporal characteristics of the effect of road network density, it is suggested to appropriately increase the road network density in Baoan District, Guangming District, and Luohu District to improve urban vitality. This proposal can serve as a reference for the local authorities and stakeholders involved in the design of road networks.

The third recommendation is to build a mixed-use city, not a "chaotic" one. Based on the results, we propose enhancing the land use mixture in the western and eastern parts of Shenzhen to promote urban vitality. However, in central regions such as Yantian District and Longgang District, it may not be feasible to improve urban vitality only by improving land use mixture. The suggestion can provide a reference for local authorities to make urban plans.

The fourth recommendation is to improve the convenience of public transport in the city. Bus stops should be added to cover all neighborhoods as much as possible. At the same time, bus routes should be optimized to reduce transfer times and improve the efficiency of urban bus operations. It is suggested that subway stations should be built in the east of Longgang District, Pingshan District, and Dapeng New District to improve the urban vitality of these areas. This proposal can serve as a reference for the local authorities and stakeholders involved in the optimization of the public transportation system.

The last recommendation is to increase the amount of CBDs in the eastern region by building a polycentric urban structure. Several CBDs in Shenzhen are mainly located in the southwest, while the east–west span of Shenzhen is wide, and the radiation of the CBD is more limited for the eastern region. Therefore, by adding commercial centers in eastern regions such as Longgang and Pingshan, the convenience of residents' lives can be enhanced and multiple regional centers can be formed, thus leading to the enhancement

of the urban vitality in the surrounding areas. The suggestion can provide a reference for local authorities and real estate developers to make CBD and commercial real estate development plans.

It is worth pointing out that the analysis unit formed by the road network segmentation used in this paper was relatively coarse, which may lead to a lack of refinement in the policy recommendations presented in this study.

### 5.3. Limitations and Prospection

There are some shortcomings in our study. First, the Baidu heat map data used in this paper only contained weekdays and weekends, thus, the time range was not long enough. In future studies, data over longer periods should be introduced to provide a more comprehensive understanding of the urban vitality in Shenzhen city. Second, our study is not comprehensive enough to characterize urban vitality only from the perspective of population activity distribution. In the future, multi-source big data such as POI and nighttime lighting can be used to research urban vitality from different perspectives. Third, the influencing factors in this paper only involved the built environment of the city, which is not comprehensive enough. In the future, indicators such as the natural environment and human economy can be taken into consideration to explore the influencing mechanisms of urban vitality more comprehensively. Forth, the TAZ units used in our study were relatively coarse to avoid generalized results. In future studies, we plan to use more fine-grained analysis units (e.g., blocks) to obtain more accurate and novel results. In addition, the effect of road type on the results was ignored in our study. As we know, road type includes the logic of accessibility, as it is associated with carrying capacity. Thus, the design of indicators for accessibility in future studies needs to be more rationalized. Last, the conclusions drawn in this paper are not necessarily appropriate for other cities. In the future, the same methodology can be applied to study multiple cities and compare the similarities and differences of the results to provide inspiration for urban vitality research.

### 6. Conclusions

The main conclusions of this paper are as follows.

(1) On both weekdays and weekends, the urban vitality in Shenzhen showed a pattern of being high in the west and south and low in the east and north, and the high-vitality areas showed a clustered, polycentric character. The distribution of high-vitality areas on weekdays did not change significantly within a day. The distribution of urban vitality on weekends was more dispersed and did not have obvious temporal variation. The temporal change in urban vitality was more dramatic during the weekdays than during the weekends.

(2) The effects of built environment factors on urban vitality exhibited significant temporal differences. Population density, building density, bus stop density, and distance to subway station showed positive effects on weekdays and weekends. Distance to the CBD exhibited a negative effect on weekdays and weekends and showed a significant diurnal difference. Road network density and functional mix, on the other hand, exhibited positive and negative effects throughout the day.

(3) In terms of spatial distribution, the effect of population density and building density on urban vitality was always positive, but the intensity of the effect exhibited differences with the TAZ units. The coefficients of road network density, functional mix, bus stop density, and distance from metro stations had both positive and negative values in the study area. Distance from the CBD mainly showed a negative impact, except for areas farther away from the CBD.

**Author Contributions:** Conceptualization, Zhitao Li and Guanwei Zhao; methodology, Zhitao Li and Guanwei Zhao; software, Zhitao Li and Guanwei Zhao; validation, Zhitao Li; formal analysis, Zhitao Li; investigation, Zhitao Li; resources, Zhitao Li; data curation, Zhitao Li and Guanwei Zhao; writing—original draft preparation, Zhitao Li and Guanwei Zhao; writing—review and editing,

Zhitao Li and Guanwei Zhao; visualization, Zhitao Li and Guanwei Zhao; supervision, Guanwei Zhao. All authors have read and agreed to the published version of the manuscript.

**Funding:** This research was funded in part by the Guangzhou Science and Technology Plan Project—Joint Project Funding by City and University (grant no. 202102010413), and Training Programs of Innovation and Entrepreneurship for Undergraduates in Guangzhou University, Guangdong Province, China (grant no. S202311078030); the National Natural Science Foundation of China (grant no. 42071443).

**Data Availability Statement:** No new data were created or analyzed in this study. Data sharing is not applicable to this article.

**Acknowledgments:** The authors would like to thank the editors and anonymous reviewers for their insightful suggestions and comments.

**Conflicts of Interest:** The authors declare no conflict of interest.

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
