# Peer review of "Revealing the Spatio-Temporal Heterogeneity of the Association between the Built Environment and Urban Vitality in Shenzhen"

_ijgi, doi:10.3390/ijgi12100433_

Round 1

Reviewer 1 Report

1 the abstract must be improved. The spatio-temporal heterogeneity of association between built environment and urban vitality is not unclear. Shenzhen is just a study area.  T

2  Literature review  should be improved. I suggest to strengthen the  critical view of the literature. 

average

Reviewer 2 Report

The main authors of conceptual basis are mentioned.

Lines 85/86 – You present a title and a subtitle in sequence. It is indicated that you insert a minimum of a paragraph between them. For example, explaining what will be the subject of the part, how it will be composed.

Lines 86/151 – you present a very good bibliographic review about the possibilities of getting spatial data, and you organize the results in table 1. But about the “social media data” it is interesting to understand that we have two possibilities associated to its use: the passive VGI (Volunteered Geographic Information) and the active one. When you write about the big data that you get from mobiles, it’s the passive mode, because people don’t know that their data is resulting in mapping (in fact, when they accept the app, there are accepting to be monitored). In active mode, they contribute in a web platform, registering points with annotations, comments, images, opinions. And this second possibility is very useful to citizens listening, because you can share and distribute a link to the VGI and people contribute to it. One of the first applications was Ushahidi VGI, but there are many others.

When writing about measuring people, please mention Goodchild (2007).

Goodchild, M. (2007). Citizens as sensors: the world of volunteered geography. GeoJournal 69, 211–221.

Lines 167/214 – There are studies about the definition of main variables in urban vitality that define it as “completeness indicators”, to reach complete streets, that are based on mobility, place and environment:

Rosa, A.A., Moura, A.C.M., Fernandes Araújo, B.M. (2022). Geodesign Teaching Experience and Alternative Urban Parameters: Using Completeness Indicators on GISColab Platform. In: Gervasi, O., Murgante, B., Misra, S., Rocha, A.M.A.C., Garau, C. (eds) Computational Science and Its Applications – ICCSA 2022 Workshops. ICCSA 2022. Lecture Notes in Computer Science, vol 13379. Springer, Cham. https://doi.org/10.1007/978-3-031-10545-6_14

Line 233 – Consider using the concept of accessibility and capillarity in a future work. You use the lines of the roads divided into tracks (a segment that interrupts in the intersection of a crossing segment) with the attribute of the type of the road (avenue, local street, and so one). The type of road is associated with the carrying capacity. The kernel density measures the capillarity (alternatives of roads) and if it is weighted by a value (according to the type of the road) it includes the logic of accessibility.

Lines 298/637 – the development of the steps in spatial analysis is very well described and the maps were very useful in the understanding of the partial results. But consider in a future work to reduce the scale of analysis. The area is a big one and you present the analysis and results aggregated in a territorial unit quite generalized. In that sense, there is a risk to arrive to results and conclusions that are not a surprise to those who know the area. So, I advise to include in the final discussions that in the future you will use a smaller unit of integration of data, to avoid generalizations. The optimum territorial unit is a block.

It can be also a question to 5.2 subtitle (Urban planning implications) and 5.3 (Limitations and prospection).

Line 421 – More that Moran index, I would include LISA analysis - Local Indicator of Spatial Autocorrelation, from Anselin (1995). ESDA (Exploratory Spatial Data Analysis). Consider it in future studies, because it is able to analyze the equity in spatial distribution.

References – I would like to suggest that you read more authors that are not from your country.

Reviewer 3 Report

Generally, the paper is well written and publishable. Some minor revisions can be made. First, the quality of figures and tables can be improved. Second, some important papers can be included and reviewed. Third, the language of the paper is rough in many places. For example, "except for areas that far away from the CBD".

Recommendations of recent papers.

Xia, C., Zhang, A., Anthony G.O. Yeh. (2022). The varying relationships between multidimensional urban form and urban vitality in Chinese megacities: Insights from a comparative analysis. Annals of the American Association of Geographers, 112(1), 141–166.

The language of this paper can be further improved.
